# New Insights into Endogenous Retrovirus-K Transcripts in Amyotrophic Lateral Sclerosis

**DOI:** 10.3390/ijms25031549

**Published:** 2024-01-26

**Authors:** Laura Moreno-Martinez, Sofía Macías-Redondo, Mark Strunk, María Isabel Guillén-Antonini, Christian Lunetta, Claudia Tarlarini, Silvana Penco, Ana Cristina Calvo, Rosario Osta, Jon Schoorlemmer

**Affiliations:** 1Laboratory of Genetics and Biochemistry (LAGENBIO), Faculty of Veterinary, University of Zaragoza, Miguel Servet 177, 50013 Zaragoza, Spain; lauramm@unizar.es (L.M.-M.); osta@unizar.es (R.O.); 2Centro de Investigación Biomédica en Red en Enfermedades Neurodegenerativas, Instituto de Salud Carlos III (CIBER-CIBERNED-ISCIII), 28029 Madrid, Spain; 3Instituto de Investigación Sanitaria Aragón (IIS Aragón), 50009 Zaragoza, Spain; 4Instituto Agroalimentario de Aragón (IA2), University of Zaragoza-CITA, C/Miguel, Servet 177, 50013 Zaragoza, Spain; 5Instituto Aragonés de Ciencias de la Salud (IACS), Centro de Investigación Biomédica de Aragón (CIBA), 50009 Zaragoza, Spain; smacias@certest.es (S.M.-R.);; 6NEMO (NEuroMuscular Omnicentre) Clinical Center, Fondazione Serena Onlus, 20162 Milan, Italy; 7Neurorehabilitation Department of Milano Institute, Istituti Clinici Scientifici Maugeri IRCCS, 20138 Milan, Italy; 8Medical Genetics Unit, Department of Laboratory Medicine, ASST Grande Ospedale Metropolitano Niguarda, 20162 Milan, Italy; claudia.tarlarini@ospedaleniguarda.it (C.T.);; 9ARAID Foundation, 50009 Zaragoza, Spain

**Keywords:** amyotrophic lateral sclerosis, endogenous retroviruses, quantitative PCR, copy-specific expression, NLRP3

## Abstract

Retroviral reverse transcriptase activity and the increased expression of human endogenous retroviruses (HERVs) are associated with amyotrophic lateral sclerosis (ALS). We were interested in confirming *HERVK* overexpression in the ALS brain, its use as an accessory diagnostic marker for ALS, and its potential interplay with neuroinflammation. Using qPCR to analyze *HERVK* expression in peripheral blood mononuclear cells (PBMCs) and in postmortem brain samples from ALS patients, no significant differences were observed between patients and control subjects. By contrast, we report alterations in the expression patterns of specific *HERVK* copies, especially in the brainstem. Out of 27 *HERVK* copies sampled, the relative expression of 17 *loci* was >1.2-fold changed in samples from ALS patients. In particular, the relative expression of two *HERVK* copies (Chr3-3 and Chr3-5) was significantly different in brainstem samples from ALS patients compared with controls. Further qPCR analysis of inflammation markers in brain samples revealed a significant increase in *NLRP3* levels, while *TNFA*, *IL6*, and *GZMB* showed slight decreases. We cannot confirm global *HERVK* overexpression in ALS, but we can report the ALS-specific overexpression of selected *HERVK* copies in the ALS brain. Our data are compatible with the requirement for better patient stratification and support the potential importance of particular *HERVK* copies in ALS.

## 1. Introduction

Amyotrophic lateral sclerosis (ALS) is a fatal neurodegenerative disease characterized by progressive degeneration and the functional loss of the upper and lower motor neurons (MNs) in the brainstem (including the medulla oblongata), spinal cord, and motor cortex [1]. As no cure is available [2], ALS leads to respiratory muscle dysfunction, paralysis, and death, usually within 3–5 years of diagnosis. While the disease is selective for motor neurons, multiple cell types, including muscle cells, astrocytes, oligodendrocytes, and microglia, are involved in its pathogenesis [3]. In 90–95% of cases, ALS is a sporadic disorder (sALS) with an unknown cause [2]. A small percentage of cases, 5–10%, are considered familial (fALS). Mutations in over 20 genes have been identified in fALS [4]. While apparently disparate, the products of many of these genes are associated with the U1 snRNP and RNA polymerase II (RNAP II) machinery [5], suggesting ALS defects in MN survival and RNA processing pathways. The exact etiology of ALS is still unclear, and a variety of cellular pathways are dysregulated in this disease [6].

ALS is a heterogeneous disease [7,8], and it is as yet unknown whether different phenotypes of ALS represent a clinical continuum or are associated with different pathophysiological mechanisms [9]. Altered inflammatory (auto)immune responses and activities are believed to contribute to the pathogenesis of ALS (reviewed in [10]). The systemic inflammation of both the peripheral and central nervous systems has been described in ALS patients, including the presence of proinflammatory monocytes [11]. In particular, increased mRNA expression of both interleukin 6 (IL6) and NLR family pyrin domain-containing 3 (NLRP3) in monocytes has been associated with progressing ALS [11]. In human ALS tissue, increased levels of several inflammasome components [12], including NLRP3, have been detected in astrocytes [13], and a meta-analysis showed increased blood levels of interleukin 1 beta (IL1B) in ALS [14]. Granzyme A (GZMA) and granzyme B (GZMB) are cell death-inducing enzymes that are released from granules in T-lymphocytes and natural killer cells in inflammatory conditions [15]. Levels of GZMA and GZMB are significantly increased in the sera of ALS patients compared with control groups [16].

ERVs (endogenous retroviruses) are the remnants of ancestral retroviral insertions into the germline. The resulting permanent retroviral integrations, termed proviruses, are transmitted vertically to offspring (reviewed in [17]). Human ERVs (HERVs) are part of the LTR retrotransposon group, exhibiting the typical structure of a provirus: *GAG*, *POL*, and *ENV* genes flanked by two LTRs. Since the initial integration events, HERV elements have become widely distributed and occupy approximately 8% of the human genome [18]. Although many copies express RNA and proteins, no infectious HERVs have been observed to date [19]. In humans, the HERVK (human endogenous retrovirus-K, HML-2 subfamily) group is considered the most recently acquired and is composed of betaretrovirus-like ERV [20]. This group comprises a range of similar but clearly distinguishable elements in the form of over one hundred proviruses and thousands of nearly full-length solitary LTRs. The anomalous expression of LINE-1, endogenous retroviruses (ERVs), and other repetitive elements has been mentioned in the context of degenerative [21,22] and autoimmune diseases, as well as in cancer [23]. Transcripts, proteins, and even retrovirus-like particles originating from HERVK proviruses have been associated with a wide variety of diseases (reviewed in [20,22]).

*HERVK* overexpression has been reported in serum [24] and the brain and cerebrospinal fluids (CSFs) of ALS patients [25,26], and reverse transcriptase activity can be found in the blood of patients [27,28]. Similarly, an analysis of eleven postmortem brain tissue samples taken from sALS patients yielded 2–3-fold higher levels of *HERVK* mRNA encoding *GAG*, *POL*, and *ENV* genes in the postmortem cortexes of patients [25]. Over the past decade, multiple studies have shown that the pathogenicity of ALS is significantly impacted by the participation of ERVs [29]. For example, the phenotype of a transgenic mouse model expressing *HERVK ENV* is reminiscent of ALS [25], and extracellularly released HERVK ENV is toxic to neuronal cells [26]. It was recently shown that HERVK ENV protein is released extracellularly in 11 out of 14 CSF samples taken from ALS patients and causes neurotoxicity both in vivo and according to in vitro neurotoxicity assays via a mechanism that involves receptor binding [26]. HERVK has also been considered a molecular target for ALS therapy in separate clinical trials [30,31]. While these results further support a role for HERVK in ALS pathogenesis in selected cases, the extent of overexpression in postmortem samples remains controversial since additional studies have been unable to confirm the association between elevated cortical HERVK mRNA levels and ALS [26,27,32].

## 2. Results

### 2.1. ALS-Associated HERVK Expression in PBMC

We first analyzed the expression of HERVK-encoded POL and ENV transcripts in the peripheral blood mononuclear cells (PBMCs) of ALS patients and compared levels with healthy subjects (non-diseased individuals, NDIs) or subjects suffering from other neuropathies (ONPs). Of the ten ALS patients analyzed (average age, 67 ± 11.16, six males and four females), eight represent cases of familial ALS (fALS) given the presence of one of the four mutations in the ALS susceptibility genes SOD1/TDP43/FUS/C9orf72. Data on the study subjects are detailed in Table 1. Both HERVK ENV and POL expression were readily observed in ALS patients (Figure 1A,B). Relative POL expression was not increased in the ALS group, as relative expression was 0.88 ± 0.21–0.27 compared with 1.0 ± 0.39–0.64 in the control group and 1.01 ± 0.13–0.15 in the ONP group. Relative ENV expression was also not increased in the ALS group, as relative expression was 0.93 ± 0.24–0.33 compared with 1.0 ± 0.33–0.51 in the control group and 1.00 ± 0.08 in the ONP group.

There was a moderate correlation (R^2^ > 0.50) between the expression of each of these genes in both the ALS samples (Figure 1C) and the combined non-ALS samples (Figure 1D), confirming the reliable detection of expression of the retroviral elements.

### 2.2. HERVK Expression in ALS Brain

Upregulated HERVK expression was previously described in the frozen brain tissue of ALS patients in [25]. In the absence of increased HERVK levels in our PBMC samples of ALS patients, we repeated our analysis based on RNA obtained from postmortem brain samples (Table 2). We analyzed the presence of HERVK ENV transcripts in brainstem, which represented the region affected in progressive patients: eight NDIs and eight ALS individuals. In addition, disease in this region accounts for most of the worst symptoms of ALS. Simultaneously, we analyzed the cortex samples of six NDI and ten ALS patients to enable a direct comparison with published data [25]. ALS patients were compared with NDIs. Similar to the PBMC results, HERVK ENV and POL expression were readily observed in the ALS patients (Figure 1E–H). No increase in ENV or POL transcripts was observed in ALS compared with the NDIs, neither in the cortex nor in the brainstem.

The presence of contaminating genomic sequences in our RNA preparations was evaluated by comparing cDNA templates treated either with or without reverse transcriptase. The PCR amplification of these templates with specific primers yielded the results depicted in Appendix A. For both genes analyzed, the levels detected were at least 10-fold higher after an RT step, and most samples were much higher, up to 10^4^-fold. These results indicate that the levels of contaminating DNA were low compared with cDNA levels. We conclude that the expression levels we measured were not influenced by contaminating DNA.

### 2.3. Analysis of Transcribed HERVK Loci

We performed next-generation sequencing (NGS) analysis to identify individual HERVK copies with altered expression in ALS. NGS analysis was performed on amplification products obtained using the specific PCR assay for HERVK ENV, which had been previously used for expression analysis (described in Figure 1 and Appendix A). Random-primed cDNAs from the PBMCs of ALS patients (*n* = 10) and controls (*n* = 10) were used as templates. Reads (70,694 ± 24,812 per sample; 25,286–136,704; Appendix A) were mapped to the human genome according to stringent criteria described in the Section 4. Once assigned to unique genomic locations, reads corresponding to the 27 HERVK ENV loci identified in the PCR simulations in silico were extracted (Appendix A). As expected, 99.85–99.99% of mapped reads corresponded to 27 loci identified in PCR simulations in silico (Appendix A). Relative frequencies were calculated as the number of reads mapping to an individual HERVK element relative to the total number of mapped reads.

The resulting data from the PBMC samples showed that the majority of reads mapped to a limited number of loci (>5% of total reads/locus), specifically to HERVK copies located on chromosomes 1 (chr1_3); 3 (chr3_2); and, to a minor extent, chromosomes 1 (chr1_3), 5 (chr5_2), 6 (chr6), 7 (chr7_1 and chr7_2), 8 (chr8_2), 12 (chr12), 19 (chr19_2), and 22 (chr22) (Figure 2A). Lower numbers of reads mapped to the remaining 16 loci, with relative frequencies ranging from 0.01 to 1.66%. We found no significant differences between ALS patients and controls in the relative frequency of reads mapping to any of these loci (Figure 2A,B). The average fold change was 1.33 ± 0.22 (Figure 3A). Assuming that the relative frequency after mapping is a measure of relative overexpression, we were unable to find copies of HERVK that were differentially expressed in the PBMCs of ALS patients.

Although we were unable to detect increased overall expression levels of HERVK in ALS-patient-derived postmortem brain samples (Figure 1), we wondered whether specific copies of HERVK might be differentially expressed in these samples. We repeated the analysis described above for PBMCs, including samples obtained from both cortex tissue and the brainstem. The resulting data show that the origin of the reads is different in the brain versus PBMCs (Figure 2A,B). Again, most brain-derived reads mapped to a limited number of loci, in this case, to HERVK copies 3–2 (>48%) and 3–4 (7.5%). In contrast to the PBMC data, we found differences between ALS patients and controls in the relative frequency of reads mapping to several loci to which lesser numbers of reads were mapped. The average relative fold changes were 0.40 ± 1.7 and 0.59 ± 3.58 in the cortex and brainstem, respectively (Figure 3A). To better survey the differences associated with ALS, the relative fold change was calculated, excluding the 10 loci that are unchanged in patients (Appendix A). The relative fold changes among the remaining 17 loci were 0.60 ± 1.2 (*p* = 0.002) and 0.97 ± 4.1 (*p* = 0.05) in the cortex and brainstem, respectively (Figure 3B).

Reads mapping to a subset of HERVK loci, including copy_chr1-1 (not shown), copy_chr3-3, copy_chr3-5, and copy_chr16-1, differed between ALS patients and NDIs (Figure 3C). The relative frequency (ranging from 0.3 to 2.95%) of reads mapping to these loci showed an increase (copies chr3-5 and chr16-1) in ALS patients compared with controls (Figure 3C) and a greater than two-fold decrease in ALS brainstems compared with NDIs (copy chr3-3). When expressed as log2 fold changes, several copies, including chr2, chr5-1, and chr16-1, displayed differential expression (increased in ALS) to varying extents (−2.42-, −1.87-, and −0.62-fold, respectively). When corrected for multi-locus analysis, significant differential expression (*p* < 0.05; FDR < 0.05) in two loci was evident (Figure 3C): a decrease in the expression of HERVK_copy_chr 3.3 (1.8-fold) and an increase in the expression of HERVK_copy_chr 3.5 (−3.4-fold). These data indicate that the relative frequency of specific HERVK copies is altered in the brains of ALS patients, even in the absence of overall HERVK overexpression (Figure 2B). Such altered frequency appears to be derived from less abundantly transcribed but potentially ALS-related HERVK loci.

### 2.4. Inflammation Markers in ALS Brains

Peripheral inflammation is associated with ALS [8], as the induction of IL6 and tumor necrosis factor alpha (TNFα) markers, among others, has been reported in the blood samples of patients [14]. A potential infiltration of activated immune cells and the local activation of microglial cells were analyzed by assessing IL6, TNFA, and GZMB expression levels using qPCR in postmortem tissue from ALS patients. The levels of all three inflammation markers trended lower in ALS patients compared with NDIs (Appendix A). While slight decreases were found for IL6 and TNFA in the brainstem, in the cortex, we observed significant decreases in both GZMB and TNFA levels in ALS patients. TNFA levels were two-fold decreased in ALS patients compared with NDIs.

We also analyzed the expression of NLRP3, an inflammasome component (Figure 4). NLRP3 mRNA levels were increased in postmortem brain tissue from ALS patients compared with NDIs (*p* = 0.048 for the brainstem; *p* < 0.001 for all brain samples combined).

## 3. Discussion

Higher levels of *HERVK* mRNA encoding *GAG*, *POL*, and *ENV* genes in postmortem brain tissue samples taken from ALS patients have been previously reported [25]. In the hope of improving the detection and diagnosis of ALS, we searched for increased expressions of *HERVK* in patients with ALS and other neuropathies. We were unable to detect increased levels of the *HERVK* transcript in PBMCs taken from ALS patients. Although immune alterations have been described in the blood of ALS patients [14], PBMCs are not primarily affected by this disease, foreshadowing the negative results we obtained. Increased levels of *HERVK* transcripts have been reported in serum [24], but it is unknown whether these reflect circulating DNA or ribonucleotides contained in microvesicles. However, the absence of increased levels of *HERVK* transcripts in ALS brains was not anticipated. Our results confirm more recent studies in which no difference in *HERVK GAG*, *POL*, or *ENV* levels was observed in brain and spinal cord samples from ALS patients and controls [33,34]. Our failure to detect increased *HERVK* expression in postmortem ALS brains may reflect the small sample sizes employed or the different cellular composition of the clinical samples analyzed. However, a similar conclusion was reached by other studies analyzing larger numbers of samples [33,34]. Therefore, the particular *HERVK* elements, or even the HERV family overexpressed, may differ between groups and patients or depend on geographical location. Given the heterogeneity found in ALS, it is as yet unknown whether different phenotypes of ALS represent a clinical continuum or are associated with different pathological mechanisms [35]. In this context, *HERVK* overexpression may well be restricted to a subgroup of patients, as already mentioned in the initial study reporting increased expression [25]. Such stratification of cases was recently reported for *HERVW* expression in psychotic spectrum patients [36] and the induction of *HERVW ENV* expression by SARS-CoV-2 [37]. The disease-associated expression of *HERV* may, therefore, be more common than previously reported.

Out of three manuscripts closely related to our study, Garson et al. [34] only addressed overall *HERVK* levels, as opposed to the identification of specific copies. After careful assessment of PCR conditions, no increases in *HERVK* expression were reported in that study. Two previous studies examined copy-specific *HERVK* expressions in ALS. These studies analyzed cortex samples [25] and/or spinal cord tissue samples [33] and did not analyze chr3-5 and chr16-1 copies, as they were based on older versions of the genome database.

Using a different approach to ours (Illumina mRNA-Seq sequencing after ribosomal RNA depletion and posterior extraction and counting of HERVK-homologous reads), slight, statistically insignificant differences (<1.15-fold) between control and ALS samples were reported in *HERVK* copies 7q34 (c7_C), 10p14 (c10_A), and 3q21.2 (K(I)) [25]. The latter corresponds to the copy we refer to as chr3-4 (Appendix A). The *loci* we found differentially expressed in the brainstem were not identified in this previous study [25]. Another recent report describes the transcriptional profiling of HML-2 based on cloning PCR products [33,34]. With respect to *HERVK ENV*, the results were derived from only a few reads, i.e., 91 for ALS cases and 53 controls. While the study reports that “significant differences in HML-2 *loci* transcriptional activities were not seen when comparing ALS and controls”, *ENV* expression from a copy named chr3q13.2_K-3 trends toward a reduction in ALS samples [33]. This is the same copy we call HERVK_copy_chr3-3 (Appendix A), whose expression was significantly reduced in brainstem samples from ALS patients (Figure 3C). It is of future interest to repeat these findings in additional samples. Future experiments using our pipeline may be improved by the use of a spiked control to improve normalization and the use of a high-fidelity Taq. A subsequent step will and need be to determine the significance of the altered expression of this *locus*. As opposed to the increased overall expression of *HERVK* in ALS clinical samples, we identified specific copies of *HERVK* that are relatively more abundantly expressed in brain tissue but not in the PBMC samples of patients. HERVK ENV protein was reportedly present in the CSFs of most ALS patients [26], as well as in the sera of ALS individuals [38]. However, the grade of CSF toxicity was not a direct function of the ENV protein concentrations measured. While this may also result from differences in post-translational modifications of these particular ENV proteins and/or the presence of non-ENV components in the CSF, it may very well be related to the specific copy of HERVK expressed.

Our results are reminiscent of a previous analysis of the RNA-seq data of ALS vs. control brain tissue samples [39]. While no global differences were reported for LTR retrotransposon subfamilies, specific copies showed altered expression in ALS. To further evaluate the overexpression of the ALS-related *HERVK* copies we identified, specific qPCR assays should be designed, validated, and applied to patient samples. In line with this, a recent study that examined the locus-specific expression of HERVs found that individuals with ALS consistently exhibited the upregulation of one specific *HERV locus*, HML6_3p21.31c, in both the motor cortex and cerebellum when compared with control subjects [40]. In addition, a recent study on the human retrotransposon called long interspersed element-1 (L1) focused on its locus-specific expression in ALS, revealing an overall decrease in intact L1 expression in two brain regions of affected individuals, with distinct clustering based on expression patterns, highlighting the importance of understanding the regulation of specific L1 subsets in these tissues and suggesting avenues for further research [41].

The *HERVK* copies identified in this study reside in chromosomes 1, 3, and 16, which also harbor the *TARDBP* (TAR DNA-binding protein or TDP-43), *CHMP2B*, and *FUS* genes, which are frequently mutated in familial cases of ALS [42]. As the distance between *HERVK* elements and ALS susceptibility genes is large, the significance of this localization is presently unclear. Several gene mutations (including *TARDBP* and *FUS*) associated with ALS converge on RNA-binding (or -processing) proteins and RNA metabolism [6,39]. As a result, the nucleocytoplasmic shuttling of RNA–protein complexes is altered, resulting in the formation of RNA granules, the loss of RNA binding (including those derived from transposable elements [43]), and defects in RNA splicing [42]. Overexpressed *HERVK* copies may produce toxic proteins relevant to ALS progression [33]. However, RNA detected in the cytosol or in endosomes is known to activate immune pathways through intracellular receptors [44]. As opposed to increased levels of *HERVK* RNAs in ALS, pathological pathways in ALS may alter the processing of *HERVK* RNAs and make them more available to these RNA-detecting pathways [45]. Such a process may depend on the sequence or structure of specific copies. A contribution of *HERVK* transcripts to ALS can now be envisaged in different ways. HERVK-encoded ERV protein may contribute to ALS-related symptoms such as the degeneration of motor neurons and motor dysfunction, as described for forced (over)expression in mouse brains [25]. In addition, we suggest that the ALS-specific *HERVK* RNAs identified in this work may contribute to the disease through the activation of innate immune pathways.

Altered inflammatory (auto)immune responses and activities also contribute to the pathogenesis of ALS [10]. Surprisingly, our results do not show increased levels of *TNFA* and *IL6* transcripts in the cortexes or brainstems of postmortem ALS brains. The observed decrease in levels of inflammation markers in ALS brains with respect to control brains could suggest that neuronal loss in the terminal stage of the disease dampens inflammation. The observation of apparently reduced inflammation is based on a very limited set of markers and awaits confirmation via additional methods, such as histological analysis in combination with specific antibody staining in an extended set of samples.

In mouse models of ALS (e.g., SOD1), DAMPs released from injured motor neurons induce microglia to acquire an M1 phenotype, which is associated with the enhanced secretion of the proinflammatory cytokines TNFα, IL1, and IL6 [10,46]. We report increased *NLRP3* expression in ALS brains (especially in the cortex; see Figure 4A). This is the first study that describes *NLRP3* levels in postmortem brain tissue. The *NLRP3* overexpression observed confirms that inflammation in the form of NLRP3 inflammasome activation may be associated with ALS, in accordance with the activation of NLRP3 inflammasome in SOD1G93A mice [47]. At present, we do not know whether other components of this inflammasome or IL1/IL18 levels are increased in the ALS brain samples analyzed.

The NLRP3 inflammasome is activated in response to a variety of agents, including bacterial RNA and toxins, K^+^ efflux, extracellular ATP, uric acid, and mitochondrial reactive oxygen species [48]. In addition, mutant forms of *TDP43* [49], which are genetically associated with ALS (and wild-type forms, to a lesser extent), can activate microglia through the NLRP3 inflammasome [50,51]. As human immunodeficiency virus type-1 single-stranded RNA activates the NLRP3 inflammasome in human microglia [52], we suggest a similar response may occur in response to *HERVK*-derived RNA. The depletion of *TDP43* from the nucleus and aggregation in the neuronal cytoplasm [49] is also associated with the deregulated expression of transposable elements in several model systems [53]. NLRP3 inflammasome activation may depend on dual signals from mutant *TDP43* and *HERVK* derived RNA. Altogether, our results are compatible with the idea that microglial inflammation in the form of inflammasome activation is a common feature of ALS.

## 4. Materials and Methods

### 4.1. Participants, Samples, and Approvals

Blood samples from patients (Table 1) were obtained under informed written consent prior to inclusion in this study, as approved by the institutional review boards of the institutional ethical committees of Niguarda Ca’ Granda Hospital, Milan, Italy. This study was conducted according to Declaration of Helsinki principles and according to Directive 2004/23/EC of the European Parliament and of the Council.

Brain samples and data from patients included in this study (patients and non-diseased controls) (Table 2) were collected, processed, and provided by Banco de Tejidos CIEN, Fundación CIEN, Instituto de Salud Carlos III, Madrid, and by Biobanco en Red de la Región de Murcia (BIOBANC-MUR), Murcia, Spain. The latter is integrated into the Spanish Biobanks Network Platform (www.redbiobancos.es (accessed on 4 October 2019)) and registered on the Registro Nacional de Biobancos with registration number B.0000859. Patients and unrelated control individuals were recruited with informed written consent, and samples were processed following standard operating procedures as approved by the local institutional review board (El Comité de Ética de la Investigación de la Comunidad de Aragón (CEICA), approval reference CEICA PI17/0025, Zaragoza, Aragón, Spain) and the Comité Científico del banco de Tejidos de la Fundación CIEN (approval reference CCS17003).

All samples were analyzed by investigators who were blinded to both the clinical conditions and identities of the patients.

### 4.2. Experimental Design

This study included a random sample of PBMCs extracted from 10 familial ALS patients selected from a previously studied group [54], matched for age and gender as far as possible (Table 1). The “other neuropathies group” (ONP group) included diseases such as myotonic dystrophy type 1 and 2, facioscapulohumeral muscular dystrophy, Becker’s muscular dystrophy, and mild myotonia. For the postmortem brain samples, a total of 11 ALS patients and 7 non-diseased controls (NDIs) were included, and samples were matched according to the anatomical region of the brain studied (Table 2). Sample sizes were based on the availability of tissues.

### 4.3. RNA Extraction, Preparation of cDNA, and qPCR

RNA extraction and purification from blood samples were described previously in [54]. RNA quality measurements for these samples are not available. For PBMC samples, random-primed cDNA was obtained from 1 μg of total extracted RNA using the High-Capacity cDNA RT kit (Applied Biosystems, Waltham, MA, USA) [54]. RNA extraction and purification from frozen brain samples were carried out as described previously for placentas [55]. No statistical differences in mean RNA quality indicator (RQI) values were encountered between ALS patients and NDI groups (Appendix A). Random-primed cDNA was prepared from 1 µg of these RNA samples using the ThermoScript™ RT-PCR System (11146-016 Thermo Fisher, Waltham, MA, USA), as described in [55]. Using serial dilutions of cDNA produced from samples with low RQI values, we established that qPCR efficiencies (Appendix A) complied with linearity and efficiency thresholds established in [56].

Gene expression of both *HERVK* and inflammation biomarkers was analyzed via reverse transcription quantitative real-time polymerase chain reaction (RT-qPCR) on a ViiA™ 7 System (Applied Biosystems, Waltham, MA, USA) under standard thermal cycling conditions. All technical details concerning qPCR are included in a document (Appendix A) according to MIQE guidelines. Glyceraldehyde 3-phosphate dehydrogenase (*GAPDH*) and ribosomal protein L19 (*RPL19*) were used as reference genes. The sequence of the primers used to amplify *HERVK* sequences has been described before in [25] and is shown in Appendix A. Commercial Taqman assays (Thermo Fisher, Waltham, MA, USA) were employed to measure *RPL19* (Hs02338565_gH), *IL6* (Hs00985639_m1), *TNFA* (Hs01113624_g1), and *NLRP3* (Hs00918082_m1). All other primers are listed in Appendix A. Analyses were performed with a SYBR Green qPCR, using the SYBR Green Master Mix (SYBR Premix Ex Taq II (Tli RNase H Plus), RR820A, Takara, Kusatsu-shi, Japan) according to the manufacturer’s protocol. For TaqMan qPCR, the Premix Ex TaqTM (Probe qPCR) Master Mix (RR390A, Takara, Kusatsu-shi, Japan) was used according to the manufacturer’s protocol. Amplification efficiency between 90% and 110% was assessed in standard curves for all primer sets (Appendix A). All reactions were carried out in triplicate, and only measurements with a standard deviation < 0.2 were considered.

The presence of contaminating genomic sequences in our RNA preparations was evaluated by comparing amplification from templates treated in parallel with or without reverse transcriptase in standard RT buffer. To compare the expression between groups, we performed the 2^−ΔΔCt^ method, as previously described [56].

### 4.4. Next-Generation Sequencing (NGS)

cDNA obtained from ALS patients and NDIs was amplified employing specific HERVK primers [25] (Appendix A) using 0.5 U Taq DNA Polymerase (D1806, Sigma-Aldrich, St. Louis, MO, USA). NGS was carried out at the Sequencing and Functional Genomics Core Facility (Servicio Científico Técnico de Secuenciación y Genómica Funcional) of the Aragon Health Sciences Institute (IACS, Zaragoza). Library preparation and sequencing were carried out using kits approved for IonTorrent technology, following procedures recommended by the relevant manufacturers. PCR products were end-repaired and purified with Agencourt AMPure XP paramagnetic beads (Beckman Coulter, Brea, CA, USA), and amplicons were quantified using the Qubit 3 fluorometer (Qubit BR dsDNA Assay, Thermo Fisher, Waltham, MA, USA). Adapters and barcodes (Ion Xpress Barcode Adapters, Thermo Fisher, Waltham, MA, USA) were added via ligation, followed by product purification with AMPure XP beads. Libraries were prepared separately for each sample using the Kapa Lib Prep Kit for Ion Torrent (Roche, Basel, Switzerland). Libraries were quantified via qPCR (Ion Library TaqMan™ Quantitation Kit, Thermo Fisher, Waltham, MA, USA), and equimolar amounts of the samples were pooled before template preparation. Emulsion PCR, template enrichment, and chip loading were carried out manually using the Ion 520&530 Kit-OT2 (Thermo Fisher, Waltham, MA, USA), and samples were sequenced on the Ion Torrent S5XL platform using an Ion 530 chip.

Quality-filtered reads (Appendix A) (exported in fastq format) were trimmed to remove primer sequences and subsequently mapped without soft clipping to the human genome (GRCh37/hg19) using TMAP version 5.2.22, applying the setting “random among best hits” for equal best hits. Only reads > 135 bp were analyzed, using (penalty) scoring as follows: match, +1; mismatch, −10; and gap, −10. Accumulated mismatch percentages were <0.75%. Subsequently, reads mapping to selected genomic *HERVK loci* were extracted and counted. Relevant *loci* were selected as those *HERVK loci* identified by BiSearch in a virtual PCR using the HERVK *ENV* primers employed throughout this study (Appendix A). As BiSearch uses the GRch 38.92 version of the Homo sapiens genome, locations were transformed into the GRCh37/hg19 version before extraction using the Ensemble assembly converter (https://www.ensembl.org/Homo_sapiens/Tools/AssemblyConverter?db=core (accessed on 11 July 2018)).

The counting reads per locus were recalculated as percentages of reads relative to the total number of reads in the sample (Appendix A). Fold change was calculated as the median percentage of reads in a group divided by the percentage in the relevant control group. Relative fold change (the difference between the experimental value and a control value) was defined as the deviation from a 1-fold change (x = fold change; if x > 1, x − 1; if x < 1, (1/x) − 1); if x = 0, relative fold change was also set at 0).

Data were further analyzed using the DESeq2 package [57], which tests for differential expression based on logarithmic fold changes. Separate raw gene-count matrixes were obtained for NDIs and for the ALS samples. The pipeline calculates log2 fold changes and provides the probability values of the *loci*, correcting *p*-values for multiple testing (false discovery rate < 0.05). A locus with less than 1 raw count in more than 50% of samples (i.e., HERV.K_copy_chr19.1) was excluded from downstream analysis.

### 4.5. Statistical Analysis

The SPSS software was used for all analyses (IBM Corp., Armonk, NY, USA, released 2013; IBM SPSS Statistics for Windows, Version 15.0). Normality was assessed with the Shapiro–Wilk test (when *n* < 50) or Kolmogorov–Smirnov test (when *n* > 50). For the assessment of the statistical significance of differences, a specific test was performed depending on normality and the number of groups of samples compared. Normally distributed data: Student’s *t*-test (2 groups) or ANOVA test (>2 groups). Non-normally distributed data: Mann–Whitney U test (2 groups) or Kruskal–Wallis test (>2 groups) were used when appropriate. Graphs were generated using SPSS, GraphPad Prism (version 8), and Microsoft Office Excel 2010.

## 5. Conclusions

Our observations fail to support the hypothesis that elevated *HERVK* expression in the brain is generally associated with sporadic ALS. However, we identified specific copies of *HERVK*, whose expression in the brain may be associated with ALS. As our study is based on a relatively small sample size, the relationship with ALS pathology and potential implications for ALS drug targeting remain to be established. Certainly, our results endorse continued research in this area with respect to patient stratification, the presence of *HERVK*-derived RNA and proteins in ALS patients and their contribution to this disease, and the potential therapeutic benefits of suppressing *HERVK* activity.

## Figures and Tables

**Figure 1 ijms-25-01549-f001:**
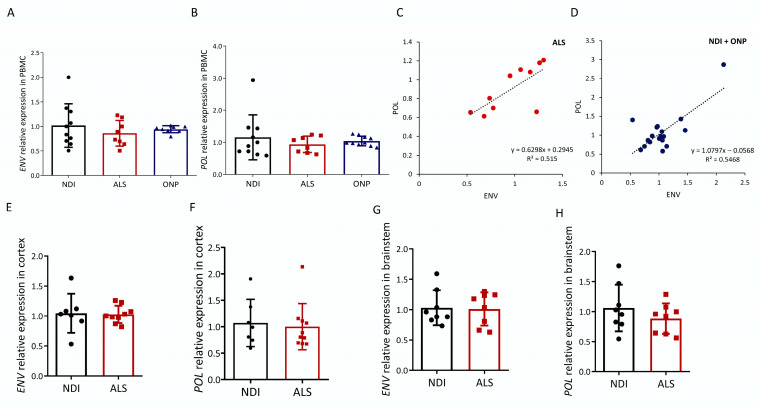
HERVK expression in ALS. Expression analysis of human endogenous retrovirus ENV and POL was carried out via qPCR in cDNA samples from ALS patients (ALS), patients with other neuropathies (ONPs), and controls (NDIs). Results were normalized using RPL19 and GAPDH as reference genes, calculated using the 2^−ΔΔCt^ method, and represented as the fold expression compared with the mean expression level in the controls. ENV (**A**) and POL (**B**) relative expression levels in PBMC samples from 10 ALS patients were compared with NDI samples (*n* = 10) and ONP samples (*n* = 10) (Student’s *t*-test; *p* = 0.982). Correlations (**C**,**D**) between mRNA expression of HERVK ENV and POL genes in the PBMC samples analyzed in (**A**,**B**) are represented, and Pearson’s correlation coefficients are indicated. ENV (**E**,**G**) and POL (**F**,**H**) levels were analyzed in the cortex and brainstem from ALS patients and control individuals (NDI). Cortex samples: ALS, *n* = 10; NDI, *n* = 6; brainstem samples: ALS, *n* = 8; NDI, *n* = 8.

**Figure 2 ijms-25-01549-f002:**
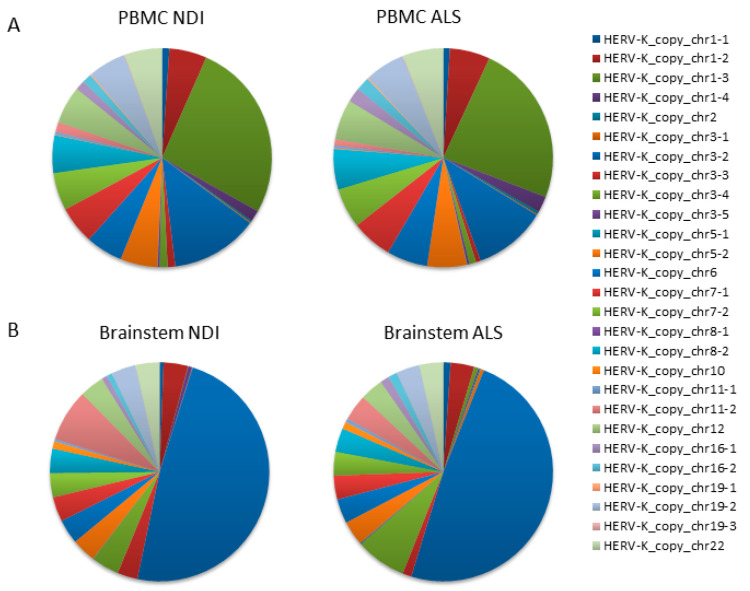
HERVK copies expressed in ALS PBMCs and the brain. A qPCR that specifically amplifies HERVK cDNA (described in Figure 1) was applied to PBMCs and brain samples taken from ALS patients and healthy controls (Table 1 and Table 2). Products were sequenced via next-generation sequencing (NGS), and the reads were mapped to the HERVK copies in the human genome. The relative frequency (%) of individual HERVK copies was calculated as the number of reads mapping to a particular HERVK locus relative to the total number of mapped reads. This relative frequency is a measure of the relative transcription of each individual HERVK copy. The 27 HERVK copies to which mapping was counted are listed on the right; their genomic location can be found in Appendix A. Relative frequencies are indicated in pie charts; color codes for each copy are indicated on the right. Analysis was performed on either the PBMCs of ALS patients (*n* = 10) and healthy controls (*n* = 10) (**A**) or on postmortem samples of the brainstems of ALS patients or non-diseased individuals (NDIs) (*n* = 4 in each group). Both the overall similarity between copies expressed in ALS vs. controls and the differentially expressed individual copies can be discerned. (**A**) Pie charts indicate the relative frequency (%) of reads mapping to each of the 27 HERVK copies in PBMC. (**B**) Same for brainstems.

**Figure 3 ijms-25-01549-f003:**
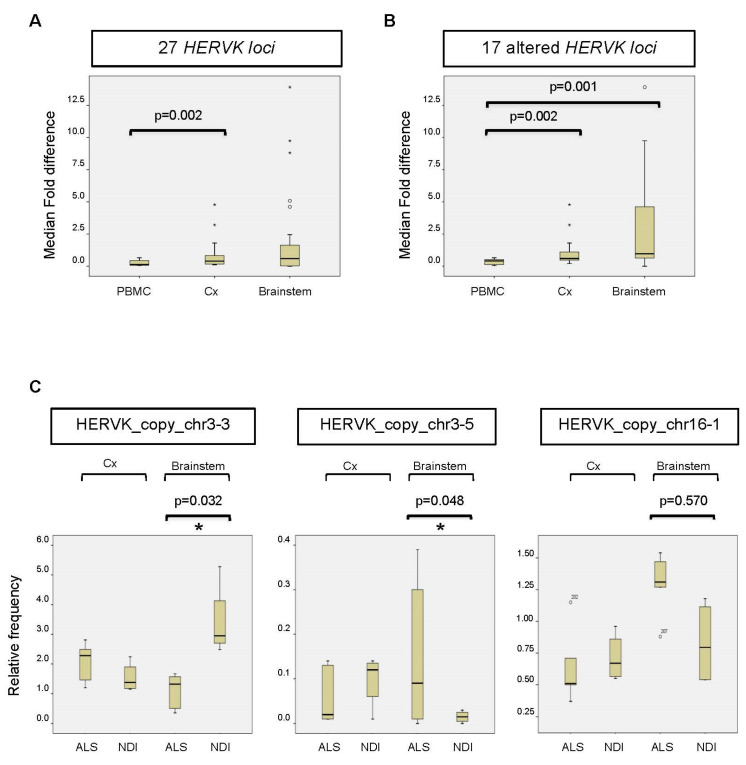
Altered expression of HERVK copies in ALS brains. (**A**) Fold change or fold difference between all 27 HERVK *loci*. The change in each locus (Δfold) was calculated as the mean/average relative frequency (%) of reads mapped to an individual HERVK *locus* in ALS patients versus NDIs (or the inverse for results < 1). Subsequently, the fold change (ΔΔfold) was calculated for each locus as Δfold-1. The median fold changes over all *loci* were calculated for PBMCs, the cortex (Cx), and the brainstem (Stem) and are represented in the figure. The corresponding *p*-values are indicated above. (**B**) Same analysis as in C for 17 loci showing differences in brains between ALS patients and NDIs. Loci that showed 0.8 < Δfold < 1.2 between ALS patients and controls for both the cortex (Cx) and brainstem were excluded from the analysis. (**C**) The relative frequency (%) of reads mapped to the individual HERVK copies is indicated. The bottom and top of the box plot are, respectively, the first and third quartiles, and the band inside the box is the second quartile (the median). Lines extending vertically from the boxes indicate variability outside the upper and lower quartiles. Outliers are indicated as individual points. IR: interquartile range. The statistical significance for the difference between ALS and NDI (calculated using the DESeq2 pipeline) is shown in the picture (* adjusted *p*-value (*p*-adj.) < 0.05).

**Figure 4 ijms-25-01549-f004:**
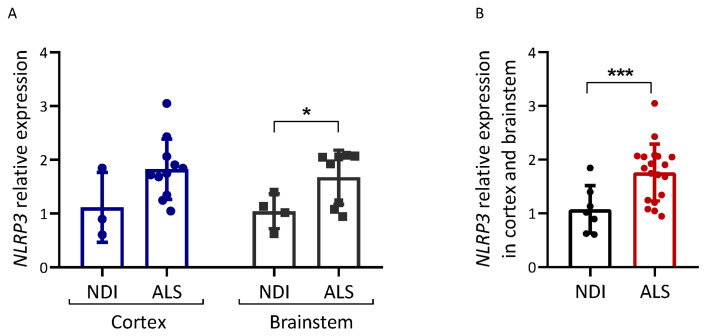
Inflammation-related gene expression in ALS brain. Expression analysis of the genes indicated was carried out via qPCR on cDNA samples prepared from the brain samples of ALS patients and controls (NDIs). Results were normalized using GAPDH and RPL19 as reference genes, calculated using the 2^−ΔΔCt^ method, and are represented as a percentage of the fold expression compared with the median expression level in the controls. NLRP3 expression was analyzed in both cortex and brainstem (**A**) and in both tissues together (**B**). ALS patient samples and control (NDI) samples were prepared from either the cerebral cortex or brainstem. Cortex ALS, *n* = 10; cortex NDI, *n* = 3; brainstem ALS, *n* = 9; brainstem NDI, *n* = 4. Student’s *t*-test: * *p* < 0.05, *** *p* < 0.001.

**Table 1 ijms-25-01549-t001:** List of PBMC samples analyzed in this study.

Patient Characteristics	fALS Patients (*n* = 10)	ONP Patients (*n* = 10)	NDIs (*n* = 10)
Gender (*n*)	6 males 4 females	6 males 4 females	4 males 6 females
Age at illness onset (mean ± SD)	60.04 ± 11.11		
Disease duration, months (mean ± SD)	34.27 ± 20.73		
Age at sampling (mean ± SD)	66.89 ± 11.61	52.37 ± 11.61	56.97 ± 10.55

NDI: non-diseased individuals group; ONP: Other neuropathies group; fALS: familial amyotrophic lateral sclerosis group (all the patients, except for two cases, carried mutations in one of the following four genes: *SOD1*/*TDP43*/*FUS*/*C9orf72*).

**Table 2 ijms-25-01549-t002:** List of brain samples used in this study.

**Patient Characteristics**	**sALS Patients** **(*n* = 10)**	**NDI** **(*n* = 9)**
Brain anatomical region (*n*)	10 frontal cortexes 8 brainstems	6 frontal cortexes 8 brainstems
Gender (*n*)	6 males 4 females	6 males 3 females
Age at sampling (mean ± SD)	58.64 ± 8.37	59.00 ± 8.73

NDI: Non-diseased individuals group; sALS: sporadic amyotrophic lateral sclerosis group.

## Data Availability

The datasets generated that form the basis of the analysis presented in the current study are available from the corresponding authors upon request.

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
