# Peer review of "New Insights into Endogenous Retrovirus-K Transcripts in Amyotrophic Lateral Sclerosis"

_ijms, 2024, doi:10.3390/ijms25031549_

Round 1

Reviewer 1 Report

Comments and Suggestions for Authors

This is a very good manuscript.

I do have some important comments for the authors to address:

1. The aims are justified quite well, but some references are fairly old. For instance, the reference that authors use to describe the impact of HERVs on diseases is from 2011. I suggest using this book chapter, as it is very comprehensive and much newer: Koks, S., & Koks, G. (2018). The Role of Human Endogenous Retroviruses (HERVs) in the Pathologies of the Nervous System. In Molecular-Genetic and Statistical Techniques for Behavioral and Neural Research (pp. 519-533). Academic Press.

2. The mechanism of how repetitive elements in the genome can cause diseases is not limited to the activation of HERVs.

3. The Ion Torrent sequencing needs more explanation. What regions were amplified, and how many primers or primer pairs were pooled into one reaction? The authors should give more details here as the amplification regions will determine their ability to detect the location of HERVKs.

4. Why was the GCRh37 reference genome used instead of GRCh38? It is not explained and not clear at all. Why lift-over was needed instead of using GCRh37 during the assay design?

Reviewer 2 Report

Comments and Suggestions for Authors

The present study evaluates a suggested biomarker for ALS which includes retroviral reverse transcriptase and the expression of HERVK- a human endogenous retrovirus element that has been associated with ALS. HERVK as a biomarker for ALS has proved controversial, basic research suggests that activation of endogenous retroviruses may recapitulate some ALS symptoms. There are ongoing clinical trials of anti-retroviral therapy in ALS that use expression of HERV-K as a secondary outcome. That study suggested a slowing of the progression of ALS but its use of historic rather than randomized controls makes interpretation complex. The exact science behind these studies is not well understood. The current study provides supportive evidence that the locus from which these RNA can be expressed can change, but does not provide evidence that the RNA is increased overall. These findings are consistent with previous studies of cohorts in the UK, and also Japan (see below). The results are at odds with two previous studies conducted in the US and one in Australia. Overall, the study makes an important point about the potential value of locus specific evaluation of HERV-K expression. Overall, this is a good contribution to the literature, that is important in its assessment and verification of a biomarker that has received substantial attention.  I felt there were some moderate limitations to this manuscript:

General

Sample size is small, - however, I felt this is consistent with other studies in this area, it also gives us a good idea of the voracity of this as a biomarker, which is seemingly not very consistent. 

Housekeeping genes, - I do not completely agree with the use of a single housekeeping gene. 

ONP - what exactly is an 'other neuropathy', does this include motor/sensory/autonomic neuropathies, or would neurodegenerative disease (AD/PD) be a better term?

Figure 2 is kind of unhelpful, its not really possible to interpret this figure.

I'm not sure the analyses of the HERVK transcripts using ion torrent is very useful, I would be concerned about the normalization method, since the library doesn't include controls. Overall, this provides a semi-quantitative ratio, but differences in depth will create many issues with this kind of analysis. Typically a spike control would be used to infer the detection sensitivity and assist with normalization, as a low number of targets violates the assumptions of DESeq. I would also strongly recommend, you use a high fidelity taq to conduct these kinds of experiments given the possibility of introducing mismatches that cause mapping to fail. Since I think it is implausible to repeat these experiments, could you please mention these limitations as caveats to your approach in the discussion.

"RNA quality measurements are not available" ??????? RNA integrity is pretty important for post-morterm samples, as these may impact detection and normalization of transcripts. 

Variance of measures:

I feel the measures of HERV-K don't necessarily evaluate all of the prior literature. The importance of HERV-K to ongoing clinical studies needs mentioning, though those findings are not 100% comparable. 

I would suggest the authors evaluate these additional studies in their discussion:

FTD / ALS cohorts - Australian Study (supportive evidence in CSF, 

https://www.nature.com/articles/s43856-021-00060-w

ALS patients -Japanese studies

https://www.sciencedirect.com/science/article/pii/S0168010222000384

Comments on the Quality of English Language

Awkward or incorrect phrasing in some of the manuscript
